# Meteorological factors and non-pharmaceutical interventions explain local differences in the spread of SARS-CoV-2 in Austria

Katharina Ledebur[1,2], Michaela Kaleta[1,2], Jiaying Chen[1,2,3], Simon D. Lindner[1,2], Caspar Matzhold[1,2], Florian Weidle[4], Christoph Wittmann[4], Katharina Habimana[5], Linda Kerschbaumer[5], Sophie Stumpfl[5], Georg Heiler[2,6], Martin Bicher[6,7], Nikolas Popper[6,7,8], Florian Bachner[5], Peter Klimek[1,2]*

**1** Medical University of Vienna, Section for Science of Complex Systems, CeMSIIS, Vienna, Austria, **2** Complexity Science Hub Vienna, Vienna, Austria, **3** Division of Insurance Medicine, Department of Clinical Neuroscience, Karolinska Institutet, Stockholm, Sweden, **4** Zentralanstalt für Meteorologie und Geodynamik, Vienna, Austria, **5** Austrian National Public Health Institute, Vienna, Austria, **6** Institute of Information Systems Engineering, TU Wien, Vienna, Austria, **7** dwh simulation services, dwh GmbH, Vienna, Austria, **8** Association for Decision Support Policy and Planning, DEXHELPP, Vienna, Austria

\* peter.klimek@meduniwien.ac.at

**Data Availability Statement:** The scripts for the model are made available at https://github.com/katharinaledebur/regional_differences_covid.

## Abstract

The drivers behind regional differences of SARS-CoV-2 spread on finer spatio-temporal scales are yet to be fully understood. Here we develop a data-driven modelling approach based on an age-structured compartmental model that compares 116 Austrian regions to a suitably chosen control set of regions to explain variations in local transmission rates through a combination of meteorological factors, non-pharmaceutical interventions and mobility. We find that more than 60% of the observed regional variations can be explained by these factors. Decreasing temperature and humidity, increasing cloudiness, precipitation and the absence of mitigation measures for public events are the strongest drivers for increased virus transmission, leading in combination to a doubling of the transmission rates compared to regions with more favourable weather. We conjecture that regions with little mitigation measures for large events that experience shifts toward unfavourable weather conditions are particularly predisposed as nucleation points for the next seasonal SARS-CoV-2 waves.

## Author summary

How weather modulates the spread of SARS-CoV-2 on fine spatio-temporal scales is still not fully understood. Here we use a controlled regional comparison to isolate the impact of five different meteorological factors, four types of non-pharmaceutical interventions as well as individual-level mobility on transmission rates in Austria. We find that more than 60% of regional variations can be explained by these factors. Temperature and humidity relate inversely with transmission rates whereas cloudiness and precipitation correlate

The data is made available at 10.5281/zenodo.6120655.

**Funding:** PK acknowledges financial support from the Vienna Science and Technology Fund WWTF under MA16-045, from the Medizinisch-Wissenschaftlichen Fonds des Buergermeisters der Bundeshauptstadt Wien, no. CoVid004, and the Austrian Science Promotion Agency FFG under 857136. The funders had no role in study design, data collection and analysis, decision to publish, or preparation of the manuscript.

**Competing interests:** The authors have declared that no competing interests exist.

with increasing transmission. We also observe a particularly strong impact of restrictions targeting large events. Our results suggest that a combination of weather shifts towards winter conditions combined with little mitigation measures for large gatherings drive the early growth of seasonal SARS-CoV-2 waves.

This is a *PLOS Computational Biology* Methods paper.

## Introduction

The SARS-CoV-2 pandemic impacted different world regions in a highly heterogeneous way. A vast body of research has sought to explain these differences by factors ranging from varying stringency of non-pharmaceutical interventions (NPIs) [1–5], socio-economic and demographic factors [6–8] to different virus variants [9–11]. Until now, only some studies paid attention to more fine-scaled spatio-temporal variations in virus spread [12–15]. Next to regional differences in NPIs, meteorological [16–21] and behavioural factors have been proposed to account for such variations [22–25]. Infection waves typically start with localized outbreaks in specific regions before case numbers start to soar on larger geographic scales [26]. Such localized outbreaks are often initiated by singular events in which transmission is dominated by a small number of individuals, [27, 28] so-called superspreading events [29]. As these events are stochastic and therefore near-impossible to predict, the question arises to which extent fine-scaled spatio-temporal variations can actually be explained or whether they are irreducibly random.

Here we seek to understand the factors that determine on finer geographic scales the degree to which the infection dynamics in one region in Austria deviated from the dynamics observed in neighboring regions. We consider factors from three different domains to account for these variations, namely meteorology (temperature, cloudiness, humidity, precipitation, wind), non-pharmaceutical interventions (targeting schools, gastronomy, healthcare, or events) as well as individual-level mobility (as inferred from telecommunications data).

Previous literature reported inconsistent associations of weather with transmission dynamics, [16–21] suggesting that the assessment of the impact of meteorological factors might depend on geography, epidemic phase and the spatio-temporal scale on which the analysis is conducted. In this work, we use meteorological forecast data derived from the mesoscale numerical weather model AROME (Application of Research to Operations at Mesoscale [30, 31]) to quantify regional weather related factors. AROME is operated by ZAMG (Zentralanstalt für Meteorologie und Geodynamik) for a domain covering the Alpine area. More details are given in section.

The literature is more consistent with regard to the transmission-reducing effects of restricted mobility [22–25]. However, such effects cannot be fully disentangled from the effects of physical distancing policies [32]. For Austria it was observed that the same regime of governmental interventions induced quite heterogeneous mobility changes across regions [33]. Here we measure mobility by means of the median Radius of Gyration of cell phone users in a region, which is a measure for the daily area travelled by these users.

In autumn 2020, Austria adopted a tiered pandemic management plan in which each region was assigned one of four alert levels (green, yellow, orange, red) signalling its

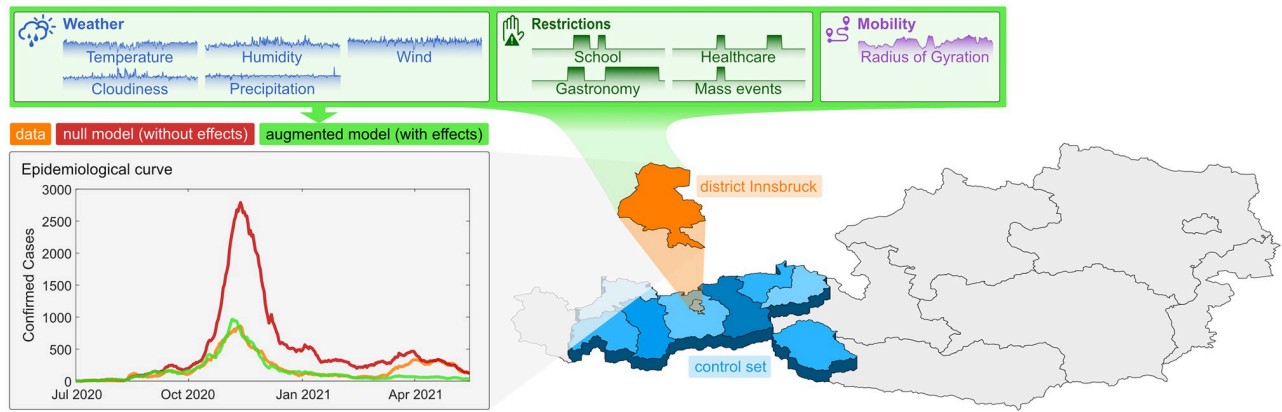

**Fig 1. Visual representation of the methodological approach.** For every one of the 116 districts in Austria the epidemiological curve of the district is compared to the epidemiological curve of the corresponding federal state. This figure shows the example of the district Innsbruck (orange) in the federal state Tyrol. The highlighted blue areas are the districts of Tyrol. More information on the districts is available on Statistics Austria. The epidemiological curves are calculated employing an age-structured compartmental model. The red curve is the epidemiological curve for Innsbruck assuming transmission rates that equal those observed in Tyrol (blue). By including a dependence of the transmission rate on weather, interventions and mobility (green), the district-independent effect sizes of these district-specific input variables can be calculated. The map layer of this figure was taken from open government data.

epidemiological risk [34]. The assessment was performed based on a set of indicators including confirmed cases, available hospital capacities and the functionality of contact tracing. Regional authorities decided on appropriate response measures in reaction to the weekly risk assessment, meaning that the same alert level could and did translate into different NPIs across regions. We used an extensive dataset on all regional response measures in Austria curated by the Austrian National Public Health Institute. We clustered the response measures into four different categories, based on whether they affect schools (bans on singing in class rooms, cloth masks when not seated, etc.), gastronomy (limits for the number of persons allowed to share a table, opening hours, etc.), healthcare settings (e.g., visitor bans), or events (size restrictions).

Our methodological approach is outlined in Fig 1. There we show a map of Austria divided into its nine federal states. The state of Tyrol (blue) is further divided into its finer administrative regions of districts. In Fig 1 we show the epidemiological curve observed in the district of Innsbruck (orange) and compare it with two models. First, we compare it to results from an age-structured compartmental model that assumes the infection rate observed in Innsbruck (orange) to be the same as the rate observed in all other districts taken together in the same federal state (blue, the control set for the orange district), giving the red epidemiological curve. Note that testing and tracing in Austria is implemented by the regional health authorities of each federal state, meaning that in our approach we always compare districts with the same strategies for testing and contact tracing. Second, we augment this model by assuming that the age-dependent transmission rates are also a function of the district-level meteorological, intervention and mobility time-series with district-independent effect sizes learned from data by means of a cross-validated nonlinear least squares solver. This approach provides us with estimates for the effect of each of the ten input time series on regional virus spread and for the degree to which the combined effects of weather, interventions and individual-level mobility can be used to explain differences in the spread between regions.

## Data and methods

### Case data

Our study period ranges from July 1 2020 to May 15 2021. The daily confirmed Covid-19 cases of 116 districts in Austria were collected through the official Austrian COVID-19 disease reporting system (EMS) [35]. A unique identification number was assigned to each positive Covid-19 test. People can be tested positive multiple times. We calculate daily numbers of positive tests per district and age group. The population size per district was available from the national statistics office and linked by the unique district ID.

The 116 districts in Austria have an average population of 77,005 (standard deviation [SD] 47,347) ranging from a minimum population of 2,000 to a maximum population of 291,134. The city of Vienna is represented as 23 individual districts with an average population of 83,520 (52,867).

### Meteorological data

We further used detailed information on the weather situation in Austria, including gridded forecast data on five meteorological parameters: total cloudiness, precipitation, 2m temperature, 2m relative humidity and 10m wind speed. The gridded data sets were provided by the mesoscale weather model AROME operated at ZAMG eight times per day producing forecasts up to 60 hours ahead. AROME is a spectral limited area model currently operating on a 2.5km horizontal grid covering the Alpine area using 90 vertical levels up to a height of approximately 35hPa. The 3D atmospheric initial conditions for the model runs are created using a variational data assimilation system (3DVAR, [36]) which is fed by a large number of observations (surface station data, radiosondes, satellite data, aircraft data, etc.). To derive the surface initial conditions, an optimum interpolation method is implemented.

For this study, the meteorological model data is provided for the study period on an hourly basis. The gridded AROME data was further aggregated to create information on a district level, where the district value of an input time series is given by the population-weighted average of the values observed in the corresponding grid cells. We calculate the daily mean values of these parameters in each district and subtract the mean value of the daily mean values of all other districts in the same federal state (control set) as input for the model.

### Data on regional NPIs

A dataset on regional Austrian NPIs was collected and curated by the Austrian National Public Health Institute. This data contains precise information on federal state, district, type and details of intervention, start and end date of registered regional measures. The data set contains 11 individual types and an additional 83 subtypes of interventions. We further grouped these types and subtypes of NPIs into 4 larger groups of restrictions: 1) restrictions in schools, 2) restrictions in gastronomy, 3) restrictions in healthcare, 4) restrictions for larger events. Intervention measures dependent on the corona traffic light were matched to the districts in appropriate traffic light colors in the given time frame. For each day in the observed time period and each district we assigned binary values of true/false for each intervention group.

### Mobility data

The mobility data contains 1,683,270 entries, including information on the date of travel, regions by postal code (which were mapped to districts), the number of devices in this particular region, and the radius of gyration over the study period. For each region only k-anonymized data is available. Any region with less than k unique devices on a specific day is

redacted. The radius of gyration is defined as the square root of the time-weighted centroid of the squared distances of the mobility events (locations) $\vec{x}_{i\mu}$ to the most prominent location (night location, center of mass) $\overline{x}_i = \frac{\sum_{\mu} \vec{x}_{i\mu} t_{i\mu}}{\sum_{\mu} t_{i\mu}}$ of each day:

$$R_{\mathrm{G},i} = \sqrt{\frac{\sum_{\tau} d(\overline{x}_i, \vec{x}_{i\tau})^2}{\sum_{\tau} t_{i\tau}}} \qquad (1)$$

Finally, as input time series for the model we consider the logarithmic radius of gyration and subtract the average logarithmic radius of gyration observed in the control set.

## Regional age-structured SIR null model without effects

To identify the impact of meteorological factors, mitigation strategies and mobility on the infection dynamics across Austria, the epidemic curves of the 116 districts of Austria are compared. We employ a parsimonious age-structured SIR model [37] to facilitate robust calibration.

Our approach makes use of the fact that when the confirmed cases per day are known, the "effective" transmission rate $\alpha$ can be calculated from data alone within the SIR model. For each district $b$, we compute a transmission rate for day $n$ and age group $a$ in the null model (without effects), $\alpha_{n,a}^0(b)$, as follows. Our null hypothesis is that the epidemiological curve of a given district $b$ (shown as orange in Fig 1) has values of $\alpha_{n,a}^0(b)$ that are identical to the transmission rate observed in all other districts of the same federal state (blue in Fig 1). We call these districts the control set of districts for $b$, $\mathcal{C}_b$. For this control set we observe the time series of cumulative confirmed cases, $C_{n,a}(\mathcal{C}_b)$. This choice of the control set is also motivated by the fact that in Austria each federal state has its own health authority that is responsible for, e.g., the testing strategy and contact tracing. Hence, with this choice of $\mathcal{C}_b$ we also control for differences in testing and tracing.

Let $S(\mathcal{C}_b)$, $I(\mathcal{C}_b)$, $R(\mathcal{C}_b)$, and $N(\mathcal{C}_b)$ be the daily numbers of susceptible, infected, recovered and total number of people in the control set, respectively. We have,

$$\alpha_{n,a}(\mathcal{C}_b) = \frac{(C_{n+1,a}(\mathcal{C}_b) - C_{n,a}(\mathcal{C}_b))N_a(\mathcal{C}_b)}{S_{n,a}(\mathcal{C}_b)\sum_{a'} c_{aa'} I_{n,a'}(\mathcal{C}_b)} \quad , \qquad (2)$$

where $c_{aa'}$ represents social mixing by age. We compute a social mixing matrix for every district to define the number of infected individuals a susceptible individual from one age group is exposed to. The social mixing matrix by age for Austria is obtained from Prem *et al.* (2017) [38] with the population being grouped into four age brackets (0–19, 20–39, 40–64 and 65 + year old individuals). We calculate a social mixing matrix for every district and federal state. To calculate the social mixing matrix entry for one of the four new age groups, we build the population-weighted sum over the corresponding social mixing matrix entries provided by Prem *et al.* (2017). The social mixing matrices for each federal state (used for the control sets) are again given by the population-weighted sum over the individual districts.

The number of susceptible, infected and removed individuals in one of the four age groups $a$, can be calculated for the control set as,

$$S_{n+1,a}(\mathcal{C}_b) \quad = \quad S_{n,a}(\mathcal{C}_b)(1 - \lambda_{n,a}(\mathcal{C}_b)) \tag{3}$$

$$I_{n+1,a}(\mathcal{C}_b) \quad = \quad I_{n,a}(\mathcal{C}_b) + \lambda_{n,a}(\mathcal{C}_b)S_{n,a}(\mathcal{C}_b) - \beta I_{n,a}(\mathcal{C}_b) \tag{4}$$

$$R_{n+1,a}(\mathcal{C}_b) \quad = \quad R_{n,a}(\mathcal{C}_b) + \beta I_{n,a}(\mathcal{C}_b) \quad , \tag{5}$$

where we defined $\lambda = \frac{\alpha_{n,a}(\mathcal{C}_b)}{N_a(\mathcal{C}_b)}\sum_{a'} c_{aa'} I_{n,a'}(\mathcal{C}_b)$. Initial conditions are given by $I_{0,a}$ as the day with the first case(s) in age group $a$ and $S_{0,a} = 1 - I_{0,a}$.

The null model (without effects) for district $b$ can then be obtained by assuming that

$$\alpha_{n,a}^0(b) = \alpha_{n,a}(\mathcal{C}_b) \ \forall \ n, a, b \quad , \tag{6}$$

holds for an age-structured SIR model for district $b$.

## Augmented regional model with effects

To quantify the impact of meteorological factors, mitigation strategies and overall mobility on the epidemic curves of all districts in Austria, we augment the null model in the following way.

In total we consider 11 input time series $X_n(b, i)$, $i = 1, \ldots, 11$. The NPI time series contain binary variables, $X_n(b, i) \in \{0, 1\}$, indicating whether a given measure was implemented in the region or not. For the meteorological and mobility time series the input signal is given by the deviation of $X_n(b, i)$ between the district and its control set, i.e., we map $X_n(b, i) \rightarrow X_n(b, i) - \langle X_n(b', i) \rangle_{b' \in \mathcal{C}_b}$ where $\langle \cdot \rangle_{b' \in \mathcal{C}_b}$ denotes the arithmetic average over all districts in the control set. To each input time series we assign an effect on the transmission rate by assuming that

$$\alpha_{n,a}(b) = \alpha_{n,a}^0(b)\Pi_{i=1}^{10}(1 + \alpha_a(i)X_n(b, i)) \ , \tag{7}$$

such that $\alpha_a(i)$ quantifies by how many percent changes in the input time series impact on the transmission rate. We further assume that $\alpha_a(i) \equiv \alpha(i)$ (i.e., effects are not age-dependent) for all input time series except NPIs related to schools, for which we assume an effect size in the first age group ($< 20y$) that is different from all other age groups. Observe that $\alpha(i)$ are district-independent effects.

## Cross-validation and effect size measurement

We solve the models for $\alpha(i)$ by means of a Levenberg-Marquandt (LM) algorithm optimizing the residual sum of squares (RSS) between age-specific incidence time series in data and model; incidence is measured as fraction of the total population. The model has one remaining hyperparameter, namely the recovery rate $\beta$. To estimate the influence of varying $\beta$, we perform a cross-validated hyperparameter search over the range $1/\beta = 4, \ldots, 35d$. We perform randomized cross-validation by splitting the time series into blocks of 28d and randomly picking 80% of these blocks for training (fitting $\alpha(i)$ via LM) and evaluating the RSS on the remaining 20% test data. To obtain the effect sizes we finally evaluate the model on the full dataset. We evaluate the model for different values of $\beta$. Different values for the recovery time ($1/\beta$) are chosen with probabilities taken from Paul *et al.* [39], namely a gamma distribution with a maximum at 19d. To measure effect sizes we consider the weighted ensemble average over models

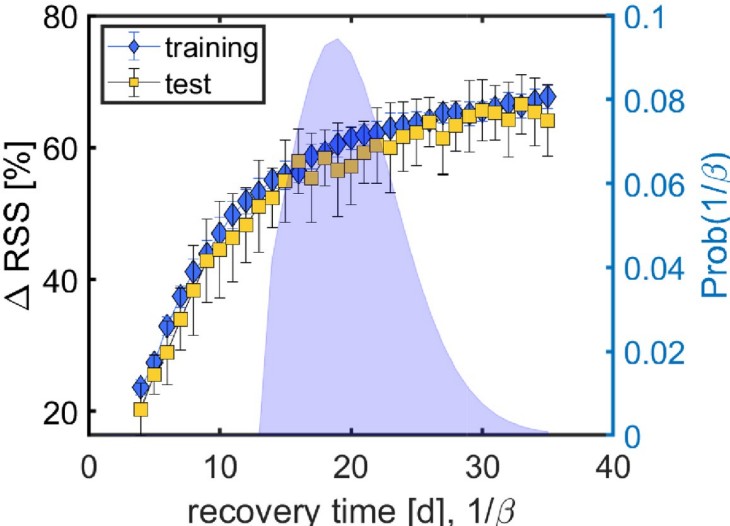

**Fig 2. Results for the cross-validated hyperparameter search.** For different recovery times, $1/\beta$, we show the percent change in RSS, $\Delta RSS$, between null and augmented model for training (blue) and test (yellow) data. For recovery times of more than 20d, the augmented model explains more than 60% of the regional variations in both test and training data. The probability distribution for $1/\beta$ [39] is shown on the axis to the right-hand side.

with different recovery times; the corresponding standard deviations are computed via weighted sample variances.

## Results

### Model calibration

For descriptive information on the input time series in the form of heat maps of the weather and mobility timeseries, see S1, S2, S3, S4, S5 and S6 Figs and S1 Table for information on the NPIs. Results for the cross-validated hyperparameter search to fix the recovery time, $\beta$, are shown in Fig 2. The larger the recovery time, the higher the percentage of regional variations that the augmented model is able to explain with respect to the null model. For recovery times of >20d, the explained variation saturates at a bit more than 60%. Values for the training data are only marginally higher than for the test data, suggesting that overfitting is not much of an issue. Fig 2 also shows the distribution of recovery times that is used to compute effect sizes over an ensemble of models with different values for $\beta$.

### Effect sizes

Results for the effect sizes of meteorological, intervention and mobility variables are shown in Fig 3. For the continuous variables (weather, mobility), see Table 1, we report the percent change in transmission rates in units of one standard deviation (SD) of the variable in the district with respect to its control set of districts. All meteorological variables show a significant effect. Strong effects can be observed for humidity, where one SD decreases the transmission rate by 15.1% (1.7), as well as for cloudiness and precipitation, where one SD increases transmission rates by 13.8% (1.5) and 15.7% (2.7) respectively. The transmission rate is also inversely associated with temperature, while wind slightly increases transmissions. Mobility, as measured by the logarithmic radius of gyration, increases transmissions by 7.0% (0.7).

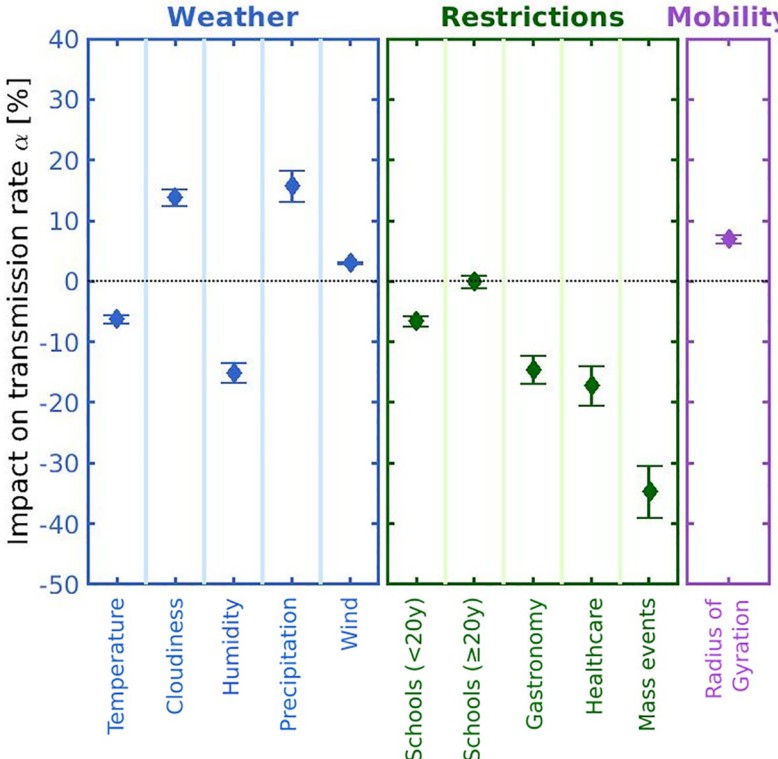

**Fig 3. Summary of effect sizes of the input variables.** Impacts on the transmission rate are shown in percent for weather variables (blue), NPIs (green) and mobility (magenta). Results for weather and mobility timeseries refer to changes in $\alpha$ for a unit change of one SD in the input. NPIs targeting large gatherings, temperature and humidity show the strongest transmission rate reductions whereas cloudiness leads to the strongest increase. Error bars denote the CI.

Considering the NPIs, restrictions targeting mass events show the strongest effect on transmissions with reductions of about 35% (4), see also Table 2. Restrictions in healthcare settings and gastronomy reduce transmissions by about 17% (3), respectively. For school measures we observe age-dependent effects. While restrictions for schools reduced transmissions by about 7% (1) in the population younger than 20y, we observe no significant impact of school measures on older age groups.

Summary of effect sizes of weather variables, NPIs and mobility is depicted for different and fixed recovery times ($\beta$ = 5, 9, 12, and 25) [S7, S8, S9 and S10 Figs].

**Table 1. Summary of effects of meteorological and mobility time series on the transmission rate.** For each variable we give its unit, the standard deviation (SD) of the input time series and the percent change with its weighted SD of the transmission rate associated with a unit SD change in the input.

| variable | unit | SD | transmission rate change [%] |
|---|---|---|---|
| temperature | ˚C | 2.4 | −6.2 (0.7) |
| cloudiness | [0–1] | 0.11 | 13.8 (1.5) |
| humidity | [0–1] | 0.063 | −15.1 (1.7) |
| precipitation | mm/h | 0.21 | 15.7 (2.7) |
| wind | m/s | 0.95 | 3.1 (0.1) |
| log. radius of gyration | $m$ | 7.9 | 7.0 (0.7) |

**Table 2. Summary of effects of NPIs on the transmission rate.** For each category of NPIs we give the number of implementations observed in our data, list typical examples of what the NPI consists of and the percent change of the transmission rate associated with a unit SD change in the input.

| category | N | examples | transmission rate change [%] |
|---|---|---|---|
| schools | 144 | cloth masks when entering schools, no indoor singing, sports only outdoors, measures to avoid mixing of school classes, ... | $< 20y$: −6.6 (0.8) <br> $\geq 20y$: −0.0 (1.0) |
| gastronomy | 123 | closing time at 10pm, limits for number of people seated at table, mandatory registration, ... | −14.5 (2.3) |
| healthcare | 161 | visitor ban or a maximum of one visitor per week, mandatory registration, FFP2 masks, ... | −17.2 (3.4) |
| events | 69 | ban or size limits of seated and unseated indoor and outdoor events | −34.6 (4.4) |

**Table 3. Result of the re-fit and re-run with variable group dropout for investigation of impact of the different variable groups on explained variation.**

| | re-fit of the model: explained variation [%] | SD [%] | re-run of the model: explained variation [%] | SD [%] |
|---|---|---|---|---|
| no dropout | 60.3 | 2.7 | 60.3 | 2.7 |
| weather dropout | 45.9 | 3.0 | 44.5 | 3.0 |
| restrictions dropout | 25.9 | 14.3 | 12.4 | 3.6 |
| mobility dropout | 60.3 | 2.8 | 58.7 | 2.9 |

## Robustness of the model

For robustness we consider adding an input time series that estimates the number of imported cases, as a variable capturing infections from the outside of the districts. However, due to their resulting small effect size, we chose to not include this variable as a main result. A summary of the effect sizes including the imported cases is depicted in S11 Fig. These small effect sizes for imported cases might suggest that most cases are imported from neighbouring regions that are already members of the control set for a district and therefore this effect is already included in the null model.

We test the impact of the variable groups weather, restrictions and mobility on the explained variation. We perform two different tests. First, we once fit the model again, setting each one of the variable groups once to zero (i.e., repeat the entire calibration procedure with a reduced set of variables). In a second robustness test, we run the model again (calibrated using all input time series), and set each variable group to zero, respectively. Again, we compute all observables in the "re-run" and "re-fit" as weighted averages using weights from [39]. From this, the effect of the different variable groups on the explained variation can be estimated, see Table 3. In both cases we find that dropping the intervention variables leads to larger decreases than dropping the meteorological data, while there is no significant reduction in RSS when dropping the mobility data.

## Discussion

In this work we aimed to understand the extent to which fine-scaled spatio-temporal variations in the spread of SARS-CoV-2 can be explained by differences in meteorological factors, NPIs, or mobility. We assume that deviations in time- and age-dependent transmission rates of a district from the transmission rate in other districts of the same federal state result from different temperatures, cloudiness, humidity, precipitation, wind, mobility, or measures targeting schools, gastronomy, healthcare or mass events. We found that taken together these factors account for more than 60% of the variation observed in regional transmission. Humidity, cloudiness and precipitation turned out to be the dominating meteorological factors, and restrictions targeting mass events showed the greatest reduction in local transmission.

As our approach aims to minimize the quadratic distance between data and model incidences, the model is optimized to explain the data in pandemic phases of high incidence. Fig 1 shows this overall tendency in that the augmented model fits the first peak (Nov 2020) substantially better than the second peak (April 2021). Austria experienced its highest incidences within the study period in most districts in November 2020, when the Austrian 7-day-incidence per 100,000 reached values of above 500. Our effect estimates are therefore particularly valid to describe the factors that might explain why certain districts were more (e.g., the district Rohrbach with a maximal 7-day-incidence of more than 1,500) or less (e.g., Gänserndorf where the incidence peaked at about 240) affected in the growth phase of this wave, i.e. the early growth behaviour of the 2020 seasonal SARS-CoV-2 wave. Note that here we do not address factors that impact the spread of SARS-CoV-2 in all districts of a federal state homogeneously, such as NPIs that where implemented on national scales, seasonal influences (as opposed to fine-scaled weather influences), or dominant virus variants.

Previous literature reported inconsistent associations of weather with transmission dynamics [16–21]. For instance, analysis of an early outbreak in China found a positive association of the infection rate with temperature, [16] as did a correlation study in Norway [17]. whereas an analysis for Spain [20] and a meta-analysis of 202 locations in 8 countries [19] observed no significant correlation. On the other hand, for the US [21] and another early, multi-city study in China [40] negative associations were reported. Also regarding the other meteorological factors the literature reports contrary results. The previously mentioned analysis of an early outbreak in China [16] reports that increase in humidity and precipitation enhances the number of confirmed cases whereas the multi-city study in China [40] reports that low humidity likely favors the transmission of Covid-19. Additionally, the correlation study in Norway [17] stated that precipitation is negatively related with Covid-19 cases. Wind may be a crucial factor in the spread of infectious diseases [41]. Several studies had examined the relation of wind speed and the transmission of coronavirus and showed heterogeneous effects in different countries [17, 42]. Nevertheless, an observational study using data from 190 countries suggested an inverse association of wind speed with transmission [43].

The reasons for this divergence in the literature regarding meteorological influences are not entirely clear. The studies mentioned above differ greatly in the included covariates, methodological approaches as well as geographic and temporal scales on which the analysis was performed, which all hinders comparability. A nonlinear dependence on temperature has also been proposed for the transmission rate [44]. In our study that adjusts for larger-scale regional trends, regional response measures and mobility we observe that increases in temperature and humidity both and independently lead to decreased transmission rates. These findings are in line with results from studies showing that higher temperature and humidity both lead to faster inactivation of SARS-CoV-2 on surfaces and in aerosols [45–47]. Airborne SARS-CoV-2 is also known to be rapidly inactivated by sunlight [48–50], in line with our finding that transmission rates increase with cloudiness.

A recent meta-analysis found that school closing was the most effective NPI in reducing the spread of SARS-CoV-2, followed by workplace closing, business closing and public event bans [51]. The school measures evaluated here were substantially less disruptive than full closures and included "soft" restrictions such as bans in indoor singing and sport activities, the requirement to wear a cloth mask when entering or leaving the school building (but not when seated in the class room), or measures to mitigate contacts between pupils from different classes. We find that such less disruptive measures nevertheless coincided with significantly reduced transmission rates in the age group $< 20y$ by about $-6.6\%$ (0.8). The different effect size of these measurements is the reason why we chose to employ an age-stratified model. If we would not stratify for age, the effectiveness of the "soft" restrictions in schools would not be visible in the

results. Our finding of a non-significant effect of school measures on transmission rates in age groups above $20y$ should not be interpreted to mean transmissions in school settings are decoupled from the population-level spread. Rather, transmission rates in older population groups are indirectly impacted by school measures in our model through the age-structured social mixing matrices.

Regarding gastronomy, we again note that our analysis focuses on less disruptive measures that did not consist of full closures, but rather of restrictions such as mandatory registration of visitors, limits for the opening hours or for the number of people seated at a table. While the effectiveness of fully closing such venues has been repeatedly established in the literature, [51] it is maybe surprising to note that less disruptive interventions also coincide with noticeably reduced transmission rates by about −14.5% (2.3).

Measures targeting the healthcare sector such as visitor bans and mandatory FFP2 masks showed a stronger effect than the less disruptive measures targeting schools and gastronomy, with reductions of−17.2% (3.4). This result emphasizes once more the necessity to protect hospitals, [52] long-term care facilities [53] and other health and social institutions [54] as one of the first lines of defense against SARS-CoV-2.

The NPI with the strongest effect size concerns public events with a reduction of −34.6 (4.4). In our dataset, this NPI also includes event bans, particularly on large unseated indoor events. While public event bans have consistently been identified as one of most effective NPIs, [51] our effect size exceeds previous estimates which are maximally in the range of −25% [4]. This difference in effect sizes can be explained by the fact that our effect estimates are susceptible to factors influencing the rapid rise in case numbers observed at the onset of the seasonal wave in autumn 2020. It is feasible that the regional onset and early growth behaviour of this seasonal wave was heavily driven by super-spreading events that have a disproportionate impact on regional transmission rates while case numbers are still relatively low. Regions with event bans already in place were not susceptible to such large increases in transmission rates in the early growth phase, which might explain our rather large effect estimate.

We find a comparably small but significant transmission-increasing effect of mobility by about 7.0% (0.7), even controlling for regional measures. While it is well-established that mobility serves as a surrogate measure to quantify the effectiveness of the corresponding NPI regime, [55] our results indicate that telecommunications data derived mobility estimates might capture additional behavioural differences.

Meteorological factors might impact SARS-CoV-2 transmission also through behavioural effects. For example, it is feasible that people prefer to meet inside rather than outside during rainy days which leads to an increase in COVID-19 cases. To the best of our knowledge, there have been no studies so far that seek to disentangle physical versus behavioural weather effects going beyond anecdotal evidence (e.g., in regions with a tropical climate high temperatures might lead people to stay indoors in air conditioned rooms, whereas in temperate climates like Austria people would be more likely to go outside). In the present study we sought to partially disentangle such effects by including mobility data as a proxy for behavioural changes (e.g., staying at home on rainy days). When dropping the mobility variable from the analysis, however, we observed no measurable decrease in explained variation. This potentially indicates that aggregate mobility is not a necessary indicator in addition to weather and NPIs to quantify such behavioural changes (should they indeed play a role in the current analysis). Further research is needed to disentangle such physical and behavioural effects across multiple climate zones.

To further understand the relative contributions of each group of variables (weather, NPIs, mobility), we evaluate how the percentage of explained variation changes when each one of the variable groups is excluded. We found that NPIs had the most impact on the explained

variation of the transmission rates followed by meteorological factors, while dropping the mobility variable showed no significant impact in these robustness tests. This finding suggests that how hard a region gets hit by an infection wave is mostly man-made—we are not helplessly exposed to seasonal forces—in the sense that control by NPIs had a larger effect on the dynamics than changes in meteorological factors.

Our work is subject to several limitations. As our parsimonious epidemiological modelling approach was designed to be easy and robust to calibrate, it does not account for incubation periods, distinguish between symptomatic and asymptomatic infections or contain undetected or quarantined cases. To make the analysis more robust with respect to potential variations in the duration of infectiousness, we performed simulations for a range of $\beta$ and calculated a weighted average of the effect sizes employing weights from Paul *et al.* [39]. The weights follow a gamma distribution with a maximum at 19d.

We also experimented with models that introduce an additional variable for the delay between input time series and their effect on transmission rates. This additional parameter did not result in substantial changes in model quality or effect sizes so we removed it per Occam's razor.

In terms of limitations we further note that our approach cannot detect spatio-temporal variations on finer scales than districts or days (e.g., whether rain is spread out over multiple hours or in a short and intense burst) and that we do not address nonlinear dependencies or interaction effects between individual input time series. In the study we use age-dependent contact patterns that have been measured before the pandemic, which could further contribute to the unexplained variation we observed when comparing model results with data. There is also a number of NPIs that was implemented much less than 50 times (e.g., regional quarantine measures) for which we could not estimate effect sizes in a statistically robust way.

Furthermore, the model does not capture infections from outside the district. However, we included time series reporting imported cases from outside the district. These imported cases do not show a significant impact on the transmission rate. See S11 Fig for the summary of effect sizes including the imported cases.

In conclusion, we find that regional differences in SARS-CoV-2 spread can in large parts be explained by a combination of meteorological factors and regional NPIs. Our approach focuses particularly on factors influencing the peak of the seasonal autumn 2020 wave in Austria, which appears to be mostly driven by differences in temperature, cloudiness, humidity, and policies targeting large public events. These findings have implications for what to expect for upcoming seasonal SARS-CoV-2 waves. In particular, based on our results we would expect the next wave to commence in regions where low mitigation measures targeting large public events combine with shifts toward unfavourable environmental conditions. If no mitigation measures for public events are in place, and precipitation, cloudiness and humidity move one SD in the direction of winter conditions, our model expects transmission rates to be more than twice as high compared to a region with control measures for public events and more favourable weather. While it has been previously noted that epidemic forecasting is like weather forecasting due to uncertainties stemming from nonlinear dynamics calibrated to noisy data streams, [56] our results suggest that epidemic forecasting to some extent *is* weather forecasting, with the added difficulty of needing to control for human behaviour.

## Supporting information

**S1 Fig. Heat map of the temperature timeseries for every district.**
(TIF)

**S2 Fig. Heat map of the cloudiness timeseries for every district.**
(TIF)

**S3 Fig. Heat map of the humidity timeseries for every district.**
(TIF)

**S4 Fig. Heat map of the precipitation timeseries for every district.**
(TIF)

**S5 Fig. Heat map of the wind timeseries for every district.**
(TIF)

**S6 Fig. Heat map of the radius of gyration timeseries for every district.**
(TIF)

**S7 Fig. Result of the effect sizes for $\beta$ = 1/5.**
(TIF)

**S8 Fig. Result of the effect sizes for $\beta$ = 1/9.**
(TIF)

**S9 Fig. Result of the effect sizes for $\beta$ = 1/12.**
(TIF)

**S10 Fig. Result of the effect sizes for $\beta$ = 1/25.**
(TIF)

**S11 Fig. Summary of effect sizes including imported cases from districts.**
(TIF)

**S1 Table. Table with informations on implemented measures.** Table showing for each restriction how many days the restriction was active and in how many different districts it occurred.
(PDF)

## Acknowledgments

We thank Wolfang Knecht for help with the figures.

## Author Contributions

**Conceptualization:** Peter Klimek.

**Data curation:** Florian Weidle, Christoph Wittmann, Katharina Habimana, Linda Kerschbaumer, Sophie Stumpfl, Georg Heiler, Martin Bicher, Nikolas Popper, Florian Bachner.

**Formal analysis:** Katharina Ledebur, Michaela Kaleta, Jiaying Chen, Simon D. Lindner, Caspar Matzhold, Peter Klimek.

**Funding acquisition:** Peter Klimek.

**Investigation:** Katharina Ledebur, Michaela Kaleta, Jiaying Chen, Simon D. Lindner, Caspar Matzhold, Florian Weidle, Christoph Wittmann, Katharina Habimana, Linda Kerschbaumer, Sophie Stumpfl, Georg Heiler, Martin Bicher, Nikolas Popper, Florian Bachner, Peter Klimek.

**Methodology:** Katharina Ledebur, Peter Klimek.

**Supervision:** Peter Klimek.

**Validation:** Katharina Ledebur, Peter Klimek.

**Writing – original draft:** Katharina Ledebur, Peter Klimek.

**Writing – review & editing:** Katharina Ledebur, Michaela Kaleta, Jiaying Chen, Simon D. Lindner, Caspar Matzhold, Florian Weidle, Christoph Wittmann, Katharina Habimana, Linda Kerschbaumer, Sophie Stumpfl, Georg Heiler, Martin Bicher, Nikolas Popper, Florian Bachner, Peter Klimek.

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
