## [Decision Letter · Decision Letter 0]

29 Nov 2021

Dear Dr Klimek,

Thank you very much for submitting your manuscript "Meteorological factors and non-pharmaceutical interventions explain local differences in the spread of SARS-CoV-2 in Austria" for consideration at PLOS Computational Biology.

As with all papers reviewed by the journal, your manuscript was reviewed by members of the editorial board and by several independent reviewers. In light of the reviews (below this email), we would like to invite the resubmission of a significantly-revised version that takes into account the reviewers' comments.

We cannot make any decision about publication until we have seen the revised manuscript and your response to the reviewers' comments. Your revised manuscript is also likely to be sent to reviewers for further evaluation.

Sincerely,

Roger Dimitri Kouyos

Associate Editor

PLOS Computational Biology

Nina Fefferman

Deputy Editor

PLOS Computational Biology

Reviewer's Responses to Questions

**Comments to the Authors:**

Reviewer #1: The authors developed an SIR model with the main goal of understanding fine spatio-temporal differences of SARS-CoV-2 transmission in the 116 districts of Austria. The authors put emphasis on meteorological factors, including wind, temperature, etc. Moreover, age-mixing and non-pharmaceutical interventions (NPI) are considered to understand differences in transmission in the districts. The model combines an impressive number of different data sources, including meteorological data, mobility data, NPI and SARS-CoV-2 cases. The model is well described and the manuscript is interesting to read.

However, as with every mathematical model, parameter choices and assumptions need to be made to perform such a study. Although the authors include a variety of different factors, of course, not all factors influencing SARS-CoV-2 transmission can be included, and some simplifications need to be made. The authors put much emphasis on the percentage of transmission variation that can be explained by meteorological factors and NPI: however, given the model simplifications and parameter choices, I am not sure whether this number (i.e. 60%) is something one should put emphasis on. To me, it is not clear, what we can really learn from the results presented here. If 60% of the variation can be explained by the factors included in the model, what about the remaining 40%? What would happen if additional factors would be included (e.g. nation-wide NPIs, etc.?).

One suggestion would be to fit the model again excluding meteorological factors and compare the results with the original model: with this, one would get an estimate how much of the variation can be explained by meteorological factors. The same strategy for the other factors.

Major comments/questions concerning parameter choices and model design:

1)Age-mixing: To me, it is not clear how age-mixing was determined. For whole Austria, the authors seem to use the matrix presented in Prem et al. However, the study by Prem et al was done before the Corona pandemic, and hence, mixing might be different from before: especially given “home office”, bans of large events, home schooling, etc., I would assume that mixing in 2020 was different compared to mixing <2017. Moreover, I do not understand ow age- mixing was calculated for the districts. More on a general note: Why is so much emphasis on age-mixing, is this really necessary for the study question of interest? I would completely drop this stratification.

2)Recovery rate: I am confused about the interpretation of this parameter. Looking at the model, beta is the rate at which individuals move from the infected department to the recovered department. Individuals in the infected department are able to infect susceptible individuals. Hence, the recovery rate is describing the number of days an infected individual is able to infect a susceptible individual. The authors fix this rate at 25 days, which is – given the isolation and quarantine for infected individuals, much too long. It might take 25 days on average to recover from Covid-19 disease, but the number of days an individual can infect others should be much lower. In line 303-305, this authors state that the recovery rate should be interpreted as the time span in which changes in NPIs or weather events might influence transmission rate: I do not see why this would be an appropriate interpretation. Looking at the model, beta clearly is the time an individual can infect a susceptible individual.

3)The authors fit the model to the number of confirmed SARS-CoV-2 cases. Ideally, these data points should be the most reliable ones. However, there were different testing strategies in Austria in these 10 months of interest. If possible, the authors should include estimates on total number of tests or negative tests as well to get a more reliable number concerning total cases. Differences in testing might also explain why the model explained the two waves not with the same precision.

4)The model does not capture infections from outside the district. This feels like a simplification, which could have a huge impact – especially for districts of the larger cities. I am wondering whether one could add an additional parameter in the infected equation to reflect additional infections, like: I + lambda*S – beta*I + gamma to model a time-varying transmission from “outside”.

5)Home schooling: I am confused by the results concerning home schooling. In Line 159, the authors mention that an effect for home schooling was only considered for age < 20y. Later on, 197-199, the authors state as one of their main results that home schooling only had an effect for age < 20y. Isn’t this result a consequence of the model design? I do not understand this point, and Line 265-268 introduce even more confusion concerning interpretation of these home schooling results.

6)I am missing a discussion about district size: Is the city of Vienna one district? Is the population of Austria evenly distributed in the districts? As the federal state is the reference for the calculations, it would be interesting to know how the districts are distributed in the federal states. In general, I am wondering whether federal state as a reference is the best choice: One could, for example, take as a reference all neighboring districts independent of the federal state – especially since weather (as opposed to NPIs) should be similar in neighboring districts.

7)Given all the uncertainties that come with the model, I am really surprised to see very narrow confidence intervals in Figure 3. How were these intervals computed? And what is the authors take on “over-fitting”?

Reviewer #2: PCOMPBIOL-D-21-01637

Meteorological factors and non-pharmaceutical interventions explain local differences in the spread of SARS-CoV-2 in Austria

Summary and comments to the authors

The above-mentioned manuscript develops a modelling approach to quantify the effect of meteorological variables and non-pharmaceutical interventions in the spread of SARS-CoV-2 in Austria. I think the paper is interesting and well written but I have the following comments:

- Figure 1 should change a bit to help readers understand the spatial context of Austria. Namely, I suggest to change the background blue of the states to white or something else, so it is clear that the blue is only Tyrol. You need to add some explanation with respect to what the larger blue regions in Tyrol represent and how they differ from districts. Can you add a map of the district boundaries with some statistics, such as population and area as a supplement?

- I would also add on a supplement some maps of the meteorological covariates, say mean over the time period, so the reader can see the extend of spatial misalignment (outcome-covariate) existing in the data.

- You mention that you use only confirmed COVID-19 cases in the model. Are these confirmed with a PCR or LFT? Does it makes sense to account for the sensitivity and specificity of these tests? How good the testing coverage is in Austria and how many cases do you expect to miss? Is this missing at random or can it introduce biases to your estimates of meteorology?

- I was wondering if you could show on a supplement how the effect estimates of the meteorological covariates or NPI change with the different \\beta. You could select the extremes and show the variation, or you could even fit all selected \\beta and create an ensemble estimate of the covariates, by combining the results, whatever you think makes more sense in this particular context.

- You mention briefly in the discussion that the meteorological factors might be confounded with behavioural aspects. This behavioural aspects that vary by country, might explain the inconsistent results of the literature. I think this merits more discussion.

- A Bayesian approach, would have resulted more natural uncertainty results, and also will have helped propagating any uncertainty that comes with the selection of \\beta.

- I would assume that the data is open available, and thus suggest the code and the data to be put on a repository, to help reproducibity of the study.

Reviewer #3: In "Meteorological factors and non-pharmaceutical interventions explain local differences in the spread of SARS-CoV-2 in Austria" authors study a critically relevant and timely problem, and they provide innovative insights that help us better understand and manage the COVID-19 pandemic in terms of a rigorous and data supported approach.

How to best contain and control the spread of COVID-19 is of outstanding and immediate importance. I have enjoyed reading this paper. I find it comprehensive and clearly written, and introducing new results that will surely inspire future research along similar lines. The main messages are brought across fully supported by the presented results, and the writing is reasonably accessible to the wider audience. For these reasons, I am in favor of revisions for PLOS Computational Biology as follows.

1) In the introduction, I am missing a couple of references where similar regional data has been studied for other countries in Europe. Perhaps not necessarily with the focus on meteorological factors, but nevertheless closely related. For a topical piece for the whole of Europe, the following reference also seems very much fitting: Towards a European strategy to address the COVID-19 pandemic, Lancet 398, 838-839 (2021).

2) A note on the robustness of the experimental data used for parameter estimation would be welcome given the still relatively varying rates available in the literature for various countries and regions of the world. A note on limitations that may not be apparent to readers as to why no robustness check were made would also be useful.

3) I also do not agree so much with the statement that less attention has been paid to more fine-scaled spatio-temporal variations in virus spread. This has in fact been studied recently also in Community lockdowns in social networks hardly mitigate epidemic spreading, New J. Phys. 23, 043039 (2021) and Socio-demographic and health factors drive the epidemic progression and should guide vaccination strategies for best COVID-19 containment, Results Phys. 26, 104433 (2021).

**Have the authors made all data and (if applicable) computational code underlying the findings in their manuscript fully available?**

Reviewer #1: None

Reviewer #2: **No: **

Reviewer #3: Yes

PLOS authors have the option to publish the peer review history of their article (what does this mean?). If published, this will include your full peer review and any attached files.

Reviewer #1: No

Reviewer #2: No

Reviewer #3: No
---

## [Decision Letter · Decision Letter 1]

28 Feb 2022

Dear Dr Klimek,

We are pleased to inform you that your manuscript 'Meteorological factors and non-pharmaceutical interventions explain local differences in the spread of SARS-CoV-2 in Austria' has been provisionally accepted for publication in PLOS Computational Biology.

Best regards,

Roger Dimitri Kouyos

Associate Editor

PLOS Computational Biology

Nina Fefferman

Deputy Editor

PLOS Computational Biology

Reviewer's Responses to Questions

**Comments to the Authors:**

Reviewer #1: The authors worked through the comments carefully and addressed a variety of uncertainties. Especially the additional robustness results, clarification for the recovery rate, and new calculation for confidence intervals improved the manuscript. As with every model, some uncertainties stay, but these are discussed in the limitations section. To me, this work is interesting from a methodological viewpoint, and it is impressive how the authors combined the large amount of different data sources. In my opinion, the actual results (i.e. quantification of impact of the different factors) should still not be over-interpreted due to many remaining simplifications of the model. With regard to this, I would like to comment on one additional aspect: I am well aware that in COVID-times, research results need to be distributed fast and the public and media long for explanations, estimations and forecasting, etc. Nevertheless, I feel that publishing the results (e.g. https://www.diepresse.com/6021911/massenevents-treiben-corona-verbreitung-staerker-an-als-vermutet) before the work goes to peer review too early, especially because newspapers tend to actually over-interpret the model estimates, without giving confidence intervals or explaining model uncertainties. Comparing different (newspaper) results of different studies could then contribute even more to the general confusion of the vast amount of COVID-related estimations published in newspapers.

Reviewer #2: The authors have implemented and addressed all my comments and concerns.

Reviewer #3: The authors have revised their manuscript comprehensively and with love to detail. I warmly recommend publication in present form.

**Have the authors made all data and (if applicable) computational code underlying the findings in their manuscript fully available?**

Reviewer #1: None

Reviewer #2: **No: **As part of addressing one of my comments, the authors said: "The availability of the data is, partly, yet to be determined by the providers. However, as soon as this is cleared the data will be provided in a repository upon completion."

Reviewer #3: Yes

PLOS authors have the option to publish the peer review history of their article (what does this mean?). If published, this will include your full peer review and any attached files.

Reviewer #1: No

Reviewer #2: No

Reviewer #3: No

---

## [Editor Report · Acceptance letter]

31 Mar 2022

PCOMPBIOL-D-21-01637R1 

Meteorological factors and non-pharmaceutical interventions explain local differences in the spread of SARS-CoV-2 in Austria

Dear Dr Klimek,

I am pleased to inform you that your manuscript has been formally accepted for publication in PLOS Computational Biology. Your manuscript is now with our production department and you will be notified of the publication date in due course.

With kind regards,

Katalin Szabo
